# Intraoperative Quantification of MDS-UPDRS Tremor Measurements Using 3D Accelerometry: A Pilot Study

**DOI:** 10.3390/jcm11092275

**Published:** 2022-04-19

**Authors:** Annemarie Smid, Jan Willem J. Elting, J. Marc C. van Dijk, Bert Otten, D. L. Marinus Oterdoom, Katalin Tamasi, Tjitske Heida, Teus van Laar, Gea Drost

**Affiliations:** 1Department of Neurosurgery, University Medical Center Groningen, University of Groningen, Hanzeplein 1, 9713 GZ Groningen, The Netherlands; j.m.c.van.dijk@umcg.nl (J.M.C.v.D.); d.l.m.oterdoom@umcg.nl (D.L.M.O.); k.tamasi@umcg.nl (K.T.); g.drost@umcg.nl (G.D.); 2Department of Neurology, University Medical Center Groningen, University of Groningen, Hanzeplein 1, 9713 GZ Groningen, The Netherlands; j.w.j.elting@umcg.nl (J.W.J.E.); t.van.laar@umcg.nl (T.v.L.); 3Expertise Center Movement Disorders Groningen, University Medical Center Groningen, Hanzeplein 1, 9713 GZ Groningen, The Netherlands; 4Center for Human Movement Sciences, University Medical Center Groningen, University of Groningen, Hanzeplein 1, 9713 GZ Groningen, The Netherlands; egbert.otten@umcg.nl; 5Department of Epidemiology, University Medical Center Groningen, University of Groningen, Hanzeplein 1, 9713 GZ Groningen, The Netherlands; 6Department of Biomedical Signals and Systems, Faculty EEMCS, TechMed Centre, University of Twente, Drienerlolaan 5, 7522 NB Enschede, The Netherlands; t.heida@utwente.nl

**Keywords:** accelerometer, Parkinson’s disease, tremor, MDS-UPDRS, quantification, intraoperative

## Abstract

The most frequently used method for evaluating tremor in Parkinson’s disease (PD) is currently the internationally standardized Movement Disorder Society—Unified PD Rating Scale (MDS-UPDRS). However, the MDS-UPDRS is associated with limitations, such as its inherent subjectivity and reliance on experienced raters. Objective motor measurements using accelerometry may overcome the shortcomings of visually scored scales. Therefore, the current study focuses on translating the MDS-UPDRS tremor tests into an objective scoring method using 3D accelerometry. An algorithm to measure and classify tremor according to MDS-UPDRS criteria is proposed. For this study, 28 PD patients undergoing neurosurgical treatment and 26 healthy control subjects were included. Both groups underwent MDS-UPDRS tests to rate tremor severity, while accelerometric measurements were performed at the index fingers. All measurements were performed in an off-medication state. Quantitative measures were calculated from the 3D acceleration data, such as tremor amplitude and area-under-the-curve of power in the 4–6 Hz range. Agreement between MDS-UPDRS tremor scores and objective accelerometric scores was investigated. The trends were consistent with the logarithmic relationship between tremor amplitude and MDS-UPDRS score reported in previous studies. The accelerometric scores showed a substantial concordance (>69.6%) with the MDS-UPDRS ratings. However, accelerometric kinetic tremor measures poorly associated with the given MDS-UPDRS scores (R^2^ < 0.3), mainly due to the noise between 4 and 6 Hz found in the healthy controls. This study shows that MDS-UDPRS tremor tests can be translated to objective accelerometric measurements. However, discrepancies were found between accelerometric kinetic tremor measures and MDS-UDPRS ratings. This technology has the potential to reduce rater dependency of MDS-UPDRS measurements and allow more objective intraoperative monitoring of tremor.

## 1. Introduction

Parkinson’s disease (PD) is the fastest growing neurodegenerative disorder globally and is accompanied by debilitating motor symptoms, such as tremor [1,2,3,4]. A Parkinsonian tremor is generally observed in the extremities and is dominant in the 4 Hz to 6 Hz frequency range [4,5,6,7]. Considering the heterogenic presentation of PD and the broad range of possible and ever-evolving therapies, accurate assessment of tremor is key for the correct treatment of PD [6,8].

The most frequently used and best validated method for evaluating PD symptoms is the internationally standardized Movement Disorder Society—Unified PD Rating Scale (MDS-UPDRS) [2,9,10,11,12]. Tremor at the hands is assessed using a total of three tremor tests: the postural tremor test, the kinetic tremor test (finger-to-nose maneuver), and the rest tremor test. During these tests, the amplitude (in centimeters) of tremor is assessed by eye. The constancy of rest tremor is assessed as the (estimated) percentage of time that rest tremor is present during the measurement. As the MDS-UPDRS is currently the best validated tool available for scoring tremor, the MDS-UPDRS is used to evaluate disease progression and therapy effect on tremor symptoms [9,10,13]. 

An established therapy in advanced PD patients is Deep Brain Stimulation (DBS) of the subthalamic nucleus (STN). During this neurosurgical treatment, leads are surgically implanted in the STN to electrically suppress motor symptoms [14,15]. Another neurosurgical option for treating tremor symptoms is a thalamotomy, where a unilateral ablation is made in the thalamus [16]. During awake thalamotomy and STN-DBS surgery, MDS-UPDRS measurements are performed to examine the clinical effect of stimulation and to optimize the lead position [17,18]. 

Unfortunately, the MDS-UPDRS is associated with limitations, such as its inherent subjectivity and reliance on experienced raters [19,20,21,22]. Thus, PD treatment decisions, e.g., permanent lead position in neurosurgery, currently depend, in part, on subjective motor assessments. Several caretakers with different levels of expertise in the MDS-UPDRS are involved with the patient throughout the caretaking process, underlining the need for a reliable and rater-independent measurement method [19,20,22]. 

Objective motor measurements might overcome the shortcomings, such as substantial inter- and intra-rater variability, of visually scored scales. However, objective measures to evaluate the clinical effect of PD therapies on motor symptoms are currently lacking in clinical practice, including the intraoperative setting, as most research is performed in laboratories [22,23,24,25]. 

Research on quantifying tremor in PD patients has been performed mainly with the use of transducers [13,26,27,28,29,30]. Tri-axial accelerometry is a subcategory of transducers that in the past decade has proven to be accurate, cost-effective, and widely available [19,20,31,32]. Accelerometers are particularly well suited for the assessment of tremor, as most relevant parameters can be calculated from the measured acceleration signal [19,31,32]. Moreover, these non-invasive sensors can easily be integrated into standard clinical tests [32]. 

The main outcome measures analyzed from accelerometric data are the dominant frequency and the amplitude of the tremor [17,29,33]. As tremor is generally a distinct sinusoidal movement, its dominant frequency can be approximated using the fast Fourier transform [33]. The frequency spectra and power spectral density (PSD) can be estimated using the Burg method, by fitting an autoregressive model to the signal through minimizing the forward and backward prediction errors [34]. These accelerometric outcome measures might be applicable to calculate objective scores based on the tremor criteria stated in the MDS-UPDRS. 

As the clinical care for patients with PD currently depends on subjective evaluations of tremor, there is a need for a reliable and rater-independent measurement method for monitoring these symptoms in PD. Therefore, this study focuses on translating the MDS-UPDRS tremor tests to an objective method to rate tremor severity using accelerometry. This study also aims to show that these measurements can be performed in an intraoperative setting, allowing direct objective input for clinical decision-making. As a first step in the process of creating a more reliable and rater-independent method, the relative agreement between accelerometry-based tremor amplitude scores and MDS-UPDRS scores are examined in this pilot study. This technology could reduce rater dependency of MDS-UPDRS measurements and allow more objective intraoperative monitoring of tremor.

## 2. Materials and Methods

This pilot study was performed at the neurosurgical department of the University Medical Center Groningen (UMCG). Exemption from the act on research involving human subjects was granted by the local medical ethics review committee (METc UMC Groningen). This study was conducted in compliance with the Helsinki Declaration for research on human beings. 

### 2.1. Subjects 

Patients with PD undergoing unilateral thalamotomy or bilateral STN-DBS surgery and healthy controls were included. The participants were screened by the researcher (A.S.) and were given oral information and a participant information letter. Written informed consent was obtained for each participant. All participants were adults, and able and willing to adhere to the study. All patient participants were diagnosed with PD according to UK Brain Bank criteria [35]. None of the healthy controls had a current diagnosis of PD or any neurological disease. Exclusion criteria were diagnosis of any form of musculoskeletal system disorders or physical disabilities, recent alcohol or drugs abuse, and psychosis or current depression. None of the participants were treated with tremor-inducing drugs during the study.

### 2.2. Materials 

Two wired tri-axial accelerometers (MMA8452Q tri-Axis, Freescale Semiconductor, Inc., Austin, TX, USA) with a range of ±2 g and a sampling rate of 200 Hz were used. The accelerometers were secured in in-house developed, non-conductive plastic cases and were attached to the index fingers with adjustable silicon straps (see Figure 1). The choice for plastic cases and silicon straps was made so that the sensors could be cleaned easily. Wired sensors were used to minimize interference with other equipment in the operating room. 

The accelerometers were connected to a Windows 7 laptop via a USB port. Accelerometry data were recorded using a measurement program built in-house in LabVIEW v. 2017 (National Instruments, Austin, TX, USA). Signal analysis was performed in MATLAB v. 2021a (MathWorks, Natick, MA, USA) and statistical analysis was performed in IBM SPSS statistics v. 25 (International Business Machines Corporation, New York, NY, USA). MDS-UPDRS items 3.15–3.18 were used for the clinical assessment of the participants [12]. 

### 2.3. Measurements 

Healthy participants were measured to determine the natural variation of noise in the 4–6 Hz frequency range and to calculate thresholds from the accelerometric data. Patients were measured while undergoing a neurosurgical procedure. All measurements were performed in an off-medication state, after a washout period of at least 12 h.

The sensors were placed at the base of the index finger of each hand during all measurements. All accelerometry measurements were performed in a supine position while the participant was awake. Accelerometry data were recorded (A.S.) while a PD expert neurologist (T.v.L.) scored the motor skill tests for both hands according to the MDS-UPDRS criteria. MDS-UPDRS tests 3.15–3.18 [12] were performed to assess postural, kinetic, and rest tremor amplitude, and constancy of rest tremor (CRT), respectively. Tests lasted ten seconds each. 

In the patient population, the set of tests described above were performed before and after lead insertion. These intraoperative tests were part of the standard clinical care given at the UMCG and did, therefore, not prolong the surgeries. 

### 2.4. Data Pre-Processing 

Raw acceleration data were converted from g units to cm/s^2^. The norm of the acceleration vector in all three directions was used for all analyses, by taking the root mean square of the resultant signal of the three axes [13,29]. A non-causal second-order high pass Butterworth filter with a cutoff frequency of 0.5 Hz was used to remove low-frequency noise and gravitational acceleration effects. To suppress digital noise and higher-order harmonics, a non-causal second-order low pass Butterworth filter of 20 Hz was applied. These filter settings were chosen with the underlying aim to not alter outcome measures or calculations, following recommendations from previous research [13,17,29]. 

The approximate cumulative numerical integral of the filtered acceleration vector was computed via the trapezoidal method over a time of ten seconds (2000 samples) to calculate the velocity (cm/s) per test. The calculated velocity vector was centered in order to correct for the constant that is added to the vector due to integration. Next, the corrected velocity was integrated numerically to the displacement (cm) of the sensor per test via the same integration method as described above. The displacement vector was filtered using a non-causal second-order high-pass Butterworth filter with a cutoff frequency of 1.2 Hz to suppress low-frequency trends caused by integration. For the kinetic tremor tests, a cut-off frequency of 3 Hz was maintained for this high pass filter, to suppress the amplitude of low frequent arm movements that occurred during this type of test. 

### 2.5. Accelerometry Measures 

To assess postural, kinetic, and rest tremor, the area under the curve (AUC) of tremor power within the 4–6 Hz tremor frequency band, and the amplitude were calculated from the accelerometry data. The periodogram PSD estimate was calculated to find the power of the acceleration norm (cm/s^2^) [29]. Next, the AUC of the power in the 4 Hz and 6 Hz frequency band (PAUC_tremor_: (cm/s^2^)^2^) was calculated from this periodogram via trapezoidal numerical integration. For the assessment of constancy of tremor, the PAUC_tremor_ was calculated for each second of the rest tremor test, resulting in ten values of PAUC_tremor_ per test. The percentage of time that tremor was present (T_tremor_: %) was calculated by summing the number of seconds where the PAUC_tremor_ was above the CRT threshold calculated from the healthy group (see Section 2.6). The amplitude of the performed test was calculated by determining the mean of all peaks in the absolute displacement vector, and multiplying this mean by two. 

### 2.6. Objective MDS-UPDRS Scores 

Following the calculation of accelerometric outcome measures, objective scores were calculated from the accelerometric data based on the criteria stated in the MDS-UPDRS. The descriptive evaluations of the MDS-UPDRS items 3.15–3.18 were applied for determining the score of the acceleration measurements via a scoring algorithm written in MATLAB. 

First, thresholds were calculated to distinguish healthy and asymptomatic measurements (MDS-UPDRS score 0) from symptomatic measurements (MDS-UPDRS score ≥ 1). As the only criteria given for an MDS-UPDRS score of 0 was “no tremor”, PAUC_tremor_ of each test was calculated from the accelerometric data of the healthy control group to determine the natural variation of noise between 4 and 6 Hz. These natural variations were used to determine thresholds that distinguish measurements of MDS-UPDRS score 0 from those of MDS-UPDRS score ≥ 1. The Kolmogorov–Smirnov test of normality (one-sample K–S test) was used to determine whether the data were normally distributed. In normally distributed data, about 95% of the values are expected to fall within two standard deviations from the population mean. This rule was used to determine the thresholds for PAUC_tremor_ of all tremor tests of the healthy controls. 

#### 2.6.1. Postural, Kinetic and Rest Tremor Amplitude 

If the PAUC_tremor_ of the test was below the calculated threshold, a score of 0 (“no tremor”) was given. Otherwise, a score of 1 or higher was given, based on the amplitude that was measured during that test. The same amplitude limits used in the MDS-UPDRS item 3.15 were applied for determining the score of the postural tremor tests measured using accelerometry, as described in Table 1. The same method was used for the kinetic and rest tremor tests. Table 1 shows the limits from the MDS-UPDRS items 3.16 and 3.17. 

#### 2.6.2. Constancy of Rest Tremor 

If the PAUC_tremor_ of the test was below the threshold of the rest tremor test (MDS-UPDRS item 3.17), a CRT score of 0 (“no tremor”) was given. Otherwise, the CRT score was based on the calculated PAUC_tremor_ per second. This variable was calculated for all healthy measurements to determine the CRT threshold. Next, it was determined if tremor was present during each second of the test in the patient population. A tremor was detected if PAUC_tremor_ per second was above the CRT threshold. Each second that tremor was present was summed up, resulting in a total number of seconds that tremor was present (T_tremor_). This value was used to calculate the percentage of the total time that tremor was present. The limits from the MDS-UPDRS item 3.18 were used, as described in Table 1. 

### 2.7. Statistical Analysis 

The association between the clinically scored MDS-UPDRS scores and accelerometric outcome measures was analyzed using linear regression. Contrast coding of the MDS-UPDRS scores was defined to be orthogonal polynomial, of which the first contrast coefficient is linear trend, to assess whether there was a linear relationship between MDS-UPDRS scores and the logarithm (log10) of the accelerometry outcome measures, as found in previous studies [31,36]. Concordance and Cohen’s weighted kappa coefficient (κ) were calculated to measure the agreement between MDS-UPDRS and accelerometry scores [37,38]. The root mean square error (RMSE) and the mean absolute error (MAE) were computed to determine how far the accelerometry scores deviated from the clinically scored MDS-UPDRS ratings.

## 3. Results

Twenty-eight patients with PD undergoing thalamotomy or STN-DBS procedures were included in this study (16 men, 12 women; age (mean ± SD): 63 ± 6.9 years). For the healthy group, 26 healthy participants were included in this study (14 men, 12 women; age (mean ± SD): 63 ± 7.3 years). In the healthy group, 156 measurements were performed. Intraoperatively, 197 measurements were performed in 28 patients. No complications occurred during the performed neurosurgical procedures. 

### 3.1. Accelerometry Measures 

The main outcome measures were the PAUC_tremor_ and amplitude of the postural, kinetic, and rest tremor tests, and PAUC_tremor_ per second and T_tremor_ of the constancy of rest tremor tests. All of these parameters were log-transformed for the regression analysis, except for T_tremor_, as this outcome measure contained values of zero. The log-transformed accelerometric measures and T_tremor_ were regressed on the MDS-UPDRS scores. There was strong evidence against the null hypothesis that the log-transformed outcome measures were not linearly related to the MDS-UPDRS scores (*p* < 0.001) for postural tremor, rest tremor, and constancy of rest tremor, see Table 2. PAUC_tremor_ and PAUC_tremor_ per second tended to increase as a logarithmic function of MDS-UDPRS scores (R^2^ > 0.735) for these tests, respectively. Conversely, no strong evidence was found against the null hypothesis that the log-transformed kinetic tremor outcome measures were not linearly related to the MDS-UPDRS scores. 

In Figure 2 and Figure 3, the quantitative outcome measures of each tremor test are plotted against the given MDS-UPDRS scores. An increase in both PAUC_tremor_ and amplitude with increasing MDS-UPDRS score can be seen for the postural tremor (Figure 2a and Figure 3a) and rest tremor amplitude (Figure 2c and Figure 3c) tests. This is also the case for the PAUC_tremor_ per second of the constancy of rest tremor tests (Figure 2d). These relationships seem to be logarithmic. For kinetic tremor, 73% of PAUC_tremor_ and 73% of amplitude values of the MDS-UPDRS groups fell within the range of the healthy group (Figure 2b and Figure 3b). 

### 3.2. Objective MDS-UPDRS Scores

The results of the one-sample K–S test showed that all required data of the healthy group were normally distributed for the threshold determination. A PAUC_tremor_ threshold of 271 (cm/s^2^)^2^ was calculated from the healthy population for the postural tremor tests. This threshold was calculated to be 6237 (cm/s^2^)^2^ and 55 (cm/s^2^)^2^ for the kinetic and rest tremor test, respectively. For the constancy of rest tremor test, a PAUC_tremor_ per second threshold of 54 (cm/s^2^)^2^ was calculated from the healthy population. The contingency tables of the MDS-UPDRS scores and the accelerometry scores (ACC) in the patient population for all tremor tests are given in heatmaps in Figure 4. 

Concordance and Cohen’s κ coefficient between the MDS-UPDRS and ACC scores per type of test, and the average error (RMSE and MAE) are given in Table 3. Concordance between the MDS-UPDRS scores and the objective scores was strong for all four tremor tests. Cohen’s κ results showed a substantial agreement for rest tremor amplitude and constancy, and postural tremor and a fair agreement for kinetic tremor tests [37,38]. As indicated by the MAE, accelerometry scores deviated on average less than half a point from the MDS-UPDRS scores.

## 4. Discussion

In this pilot study, MDS-UPDRS tremor assessments were translated to an objective method using accelerometry measurements. The relationship between clinical MDS-UPDRS scores and the quantitative measures calculated from accelerometry data was investigated, as well as the relative agreement between clinically scored MDS-UPDRS ratings and objective scores calculated from accelerometric data. A quantitative scoring algorithm for MDS-UPDRS tests 3.15–3.18 based on accelerometry measurements was proposed. 

This study showed that accelerometric measurements are a feasible option for quantifying the MDS-UPDRS tremor tests. The trends were consistent with the logarithmic relationship between tremor amplitude and MDS-UPDRS score reported in previous studies [13,31,36]. This study also showed discrepancies between accelerometric outcome measures and clinically scored MDS-UPDRS ratings, mainly in the kinetic tremor tests. 

### 4.1. Contributions 

The main contribution of this study is translating MDS-UPDRS tremor measurements to an objective scoring algorithm based on accelerometry. Healthy controls were included in this research to calculate natural variation of PAUC_tremor_ between 4 and 6 Hz, and to provide baseline thresholds for the objective scoring algorithm. As PD treatment decisions during awake thalamotomy and STN-DBS surgery, e.g., permanent lead position, currently depend, in part, on subjective MDS-UPDRS tremor assessments, this study focused on neurosurgical PD patients. In this study, the quantification of tremor measurements was investigated in the intraoperative setting, providing direct objective input for clinical decision-making during neurosurgery. The results provided by the algorithm are clinically meaningful and easy to interpret, as they are based on the same MDS-UPDRS criteria PD caregivers are used to working with.

### 4.2. Accelerometry Measures 

The results of the intraoperative use of our accelerometry-based scoring algorithm were presented. The PAUC_tremor_ and amplitude outcomes of the tremor measurements in both the healthy and the patient group were consistent with previous research, namely a logarithmic relationship with the given MDS-UPDRS tremor scores [13,31,36]. 

Another main finding was the relatively large variation in PAUC_tremor_ during the kinetic tremor tests in the healthy group. This led to a large overlap between this outcome measure of patients with PD and healthy subjects (Figure 2c). The power in the frequency band of 4–6 Hz is possibly influenced by the higher harmonics of the intentional kinetic arm movement made during these tests. Furthermore, when participants move their arm too fast, the frequency of the arm movement can overlap with that of the tremor frequency band, resulting in a higher measured amplitude and PAUC_tremor_ between 4 Hz and 6 Hz.

Overall, it stood out that relatively few patients were included that scored 2, 3, or 4 on the MDS-UPDRS scale. This was most likely caused by the clinical effect of the surgery, as only the measurements performed before lead placement received high MDS-UPDRS scores. More than half of all tests were performed after lead insertion, and were influenced by the microlesion effect, resulting in lower MDS-UPDRS scores for post insertion measurements [18]. 

In this study, the constancy of rest tremor was calculated over the time that the rest tremor test was performed. As a next step, rest tremor constancy should be assessed during the whole examination. 

### 4.3. Objective MDS-UPDRS Scores 

Strong concordance was found between the clinically scored MDS-UPDRS ratings and accelerometry-based scores, whereas Cohen’s κ was considerably lower. This could be explained by the large number of scores of zero (meaning no tremor was observed) in this study population. The effect of the outliers in this dataset can be seen in the high value of RMSE, indicating an error of 0.5 to 0.7 point between MDS-UPDRS and accelerometric scores. From Figure 4 one could argue that the accelerometric approach underestimates the amplitude of tremor, or that the clinical assessment overestimates it. Some studies have already shown that recorded accelerometric data and visual evaluations according to clinical scales do not correlate very well, mainly because minor changes in tremor are hard to assess by eye [17]. This is especially the case when the amplitude at baseline is small; whereas accelerometers can detect these small changes [17]. An explanation for the imperfect agreement with clinical assessments found in other studies might be that objective measurements outperform clinical scales at the level of the individual patient [23]. After all, movement sensors are able to capture subtle changes in PD symptoms as the disease progresses that cannot be captured using discrete scoring schemes, such as the MDS-UPDRS [39]. 

The large differences between the three calculated PAUC_tremor_ thresholds (Section 3.2) can be explained by the hand positions during tremor tests. Due to muscle tension that emerges when stretching the arms out in front of the body, the threshold of the PAUC_tremor_ is higher for postural tremor than for rest tremor, during which there is low muscle tension present in healthy persons. The kinetic tremor threshold is even higher due to increased muscle tension and arm movement. The large variation in PAUC_tremor_ in kinetic tremor tests possibly led to a threshold that could not sufficiently distinguish between healthy and symptomatic measurements. 

Lastly, the discrepancies between Figure 3 and Figure 4 can be explained by how the scoring algorithm first analyses whether the PAUC_tremor_ between 4 and 6 Hz of a test is above the threshold. If this is not the case, a score of 0 (“no tremor”) is given. Although the total amplitude or T_tremor_ for that test might have been substantial, these values were not taken into account for the scoring if the PAUC_tremor_ was below the threshold. This approach was chosen, as the total amplitude could also be caused by movements outside the 4–6 Hz frequency range. 

### 4.4. Inconsistencies 

One of the largest limitations in this research was insufficient measurements in patients with PD that scored above 2 on the MDS-UPDRS scale. This led to an underrepresentation of more severely affected PD patients. 

Secondly, although the scoring neurologist (T.v.L.) in this study is a PD expert, there was only one rater. To increase the reliability of the MDS-UPDRS ratings, the scores of several experienced neurologists rating the same patient should be included in future research. This will also allow for the comparison of the MDS-UPDRS inter-rater variability and that of the accelerometry scoring algorithm. 

MDS-UPDRS tests do not call for the use of instrumental aids, such as rulers, but instead rely on the naked eye. This might have caused discrepancies between the outcomes of the accelerometric measurements and the clinically scored MDS-UPDRS ratings. Previous studies have shown that the distances measured using objective techniques often do not agree with the distances that were visually estimated by the MDS-UPDRS rater [31,40].

In contrast to the PD group, the healthy group was not assessed in an intraoperative setting. This might have affected the calculated thresholds, and, therefore, possibly influenced the results of the algorithm in the PD group. The environmental effects and stress that patients undergoing surgery may experience were not simulated for the healthy controls, which might have influenced the occurrence of tremor in the healthy group. In this study, the PAUC_tremor_ values of healthy persons were used to correct for activity in the 4–6 Hz frequency band that is present when a person does not suffer from any range of Parkinsonian tremor. Still, it cannot be assumed that all power in the 4–6 Hz frequency range is caused by tremor. Objective criteria for assessing whether a spectral peak is present are essential for determining if the observed movement is tremor [7,29,36]. A dominant spectral peak between 4 Hz and 6 Hz that could be identified upon visual or automatic inspection of the periodogram should also result in a substantially higher value of PAUC_tremor_. This will be investigated further in our ongoing research.

Considering the large natural variation found in this pilot study, it is recommended to perform future studies with more healthy participants that are age-matched to the patient population. It is also recommended to include more PD participants, especially patients that score higher than a score of 1 according to the MDS-UPDRS tremor criteria. 

### 4.5. Future Perspectives 

Research on techniques for PD motor quantification and defining objective outcome measures is currently on the rise [23]. It is expected that, over time, the use of objective measurement tools will be preferred by physicians as an addition to the internationally standardized clinical scales that are currently used to assess the course of PD symptoms [41]. As stated by Elble and Ondo, transducers, such as accelerometers, can complement clinical assessment to quantify tremor severity [42]. The clinical importance of this objective method lies in the reduction of inter- and intra-rater variability, allowing clinicians to rate patients more uniformly. This will be further investigated in our ongoing research following this pilot study, where the variability and reproducibility of both the MDS-UDPRS tests and the accelerometry-based method are examined and compared. Reduced inter- and intra-rater variability in monitoring tremor will contribute to tracking the disease progression and treatment effect more precisely. To aid clinicians in optimizing patient outcomes, it is critical to present these complex quantitative data in a quickly interpretable manner, for example using scores according to the MDS-UPDRS. The proposed technique might also form a solution to the clinical problem that physicians are faced with when visually scoring dystonic and essential tremor. The frequency range and settings used in current scoring algorithm could be customized to standardized dystonic and essential tremor rating scale criteria with few adjustments [7,43,44]. 

Another promising line of research involves machine learning methods, such as neural networks or support vector machine classification of PD and healthy subjects based on accelerometry measures. Besides efficiently handling vast amounts of high-dimensional data, machine learning methods hold promise not just in early disease detection and (differential) diagnosis, but also in estimating tremor severity [30,31,39,45,46].

Future steps to take in the development of the techniques proposed in this article are data collection in “free-living” conditions, relevant to the patient’s functioning, and using this data for generating individualized feedback to clinicians, caregivers, and patients [23]. This will aid in tailoring treatments to the individual patient and increasing patient self-management.

## 5. Conclusions

This study showed that the MDS-UPDRS tremor tests can be quantified using accelerometric measurements. Substantial agreement was found between the clinically scored MDS-UPDRS ratings and accelerometry-based scores. The postural tremor, rest tremor, and constancy of rest tremor data were consistent with a logarithmic relationship reported in previous studies. Discrepancies were found between objective outcome measures and the MDS-UPDRS ratings of the kinetic tremor test, mainly due to the natural variation found in the healthy group. Future research should focus on investigating the variability and reproducibility of MDS-UPDRS measures and accelerometric outcome measures, especially for kinetic tremor. This technology could reduce rater dependency of MDS-UPDRS measurements and allow more objective intraoperative monitoring of tremor.

## Figures and Tables

**Figure 1 jcm-11-02275-f001:**
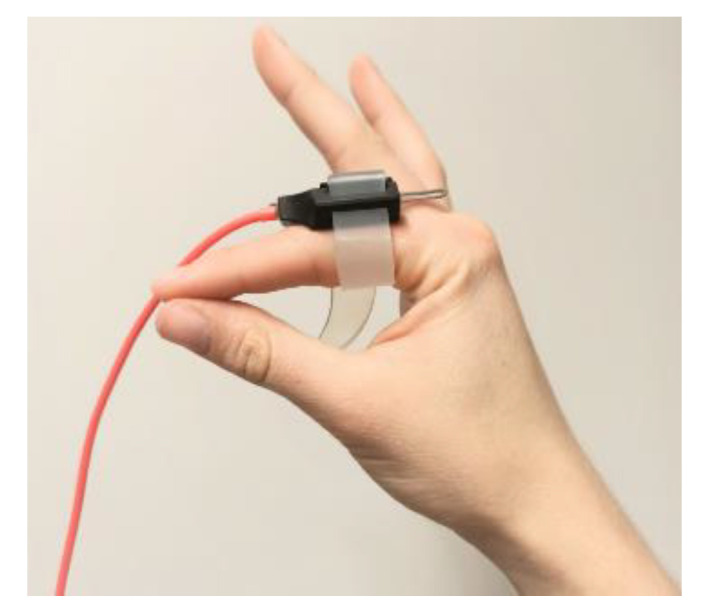
Accelerometer positioned at the base of the right index finger.

**Figure 2 jcm-11-02275-f002:**
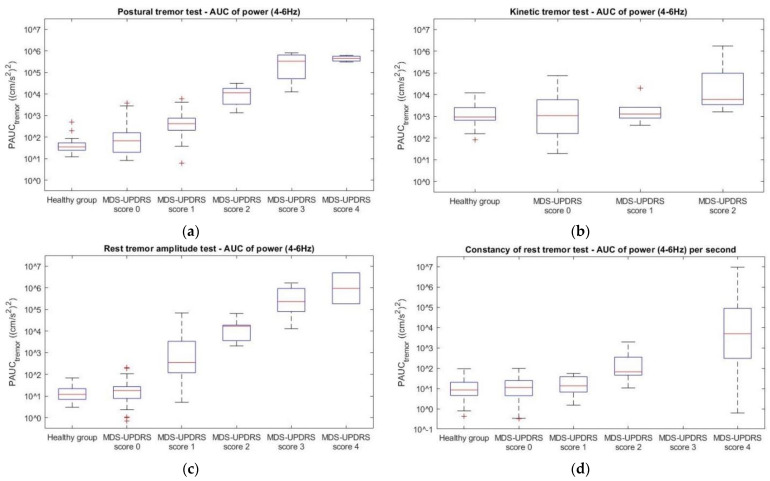
Boxplots of the PAUC_tremor_ between 4–6 Hz of the postural (**a**), kinetic (**b**), and rest tremor (**c**) tests in the healthy group and the patient population (per MDS-UPDRS score). The PAUC_tremor_ between 4–6 Hz per second of the rest tremor test in all groups is given in boxplot (**d**).

**Figure 3 jcm-11-02275-f003:**
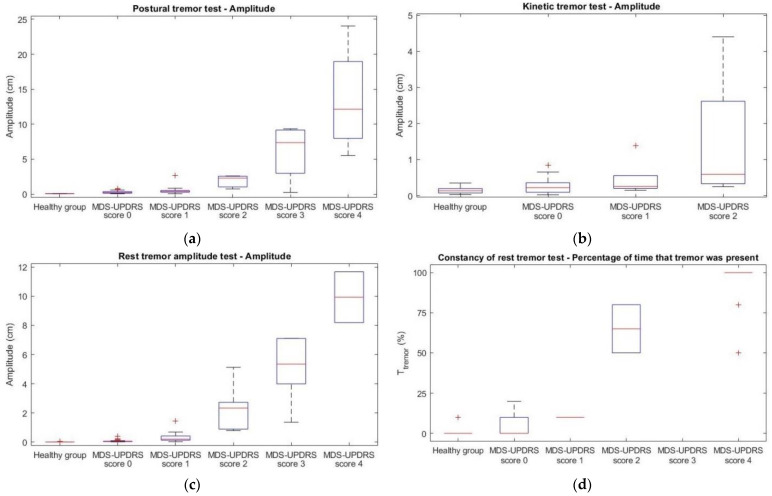
Boxplots of the total amplitude of the postural (**a**), kinetic (**b**), and rest tremor (**c**) tests, and of the T_tremor_ of the constancy of rest tremor test (**d**) in the healthy group and the patient population (per MDS-UPDRS score).

**Figure 4 jcm-11-02275-f004:**
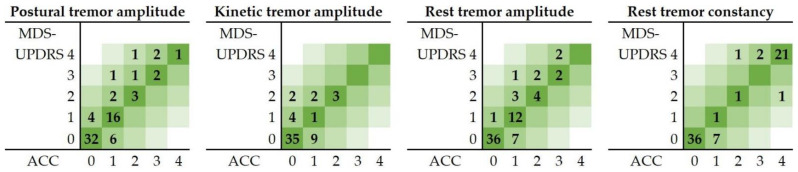
Contingency tables of the given MDS-UPDRS scores and calculated accelerometry (ACC) scores of the tremor tests in the patient population.

**Table 1 jcm-11-02275-t001:** Description of MDS-UPDRS scoring criteria [12].

Score	Scoring Criteria MDS-UPDRS Items 3.15–3.17	Score	Scoring Criteria MDS-UPDRS Item 3.18
0	No tremor	0	No tremor
1	≤1 cm in maximal amplitude	1	Tremor is present ≤25% of the time
2	>1 cm but <3 cm in maximal amplitude	2	Tremor is present 26–50% of the time
3	3–10 cm in maximal amplitude	3	Tremor is present 51–75% of the time
4	>10 cm in maximal amplitude	4	Tremor is present >75% of the time

**Table 2 jcm-11-02275-t002:** Regression analysis results.

MDS-UPDRS Test	Outcome Measure	R	R^2^ *	Coefficient **	95%CI	*p*
3.15 Postural tremor	Log(PAUC_tremor_)	0.869	0.755	3.171	2.737, 3.605	<0.001
	Log(Amplitude)	0.805	0.648	1.312	1.079, 1.544	<0.001
3.16 Kinetic tremor	Log(PAUC_tremor_)	0.379	0.144	1.749	0.584, 2.914	0.004
	Log(Amplitude)	0.513	0.264	1.030	0.560, 1.499	<0.001
3.17 Rest tremor amplitude	Log(PAUC_tremor_)	0.904	0.818	4.256	3.769, 3.769	<0.001
	Log(Amplitude)	0.912	0.832	2.032	1.811, 2.253	<0.001
3.18 Rest tremor constancy	Log(PAUC_tremor_/s)	0.857	0.735	2.322	1.985, 2.660	<0.001
	T_tremor_	0.974	0.949	7.180	6.779, 7.581	<0.001

* Coefficient of determination; ** Contrast coefficient testing for linear trend.

**Table 3 jcm-11-02275-t003:** Agreement results.

MDS-UPDRS Test	Concordance	Cohen’s κ	95%CI	RMSE	MAE
3.15 Postural tremor	76.1%	0.614	0.563, 0.665	0.569	0.268
3.16 Kinetic tremor	69.6%	0.239	0.137, 0.341	0.641	0.339
3.17 Rest tremor amplitude	77.1%	0.620	0.597, 0.643	0.521	0.243
3.18 Rest tremor constancy	84.3%	0.726	0.628, 0.824	0.493	0.186

## Data Availability

The data may be available at a reasonable request.

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
