# Peer review of "Intraoperative Quantification of MDS-UPDRS Tremor Measurements Using 3D Accelerometry: A Pilot Study"

_jcm, 2022, doi:10.3390/jcm11092275_

Round 1

Reviewer 1 Report

In this paper, the authors propose a quantitative way of determining the MDS-UPDRS for tremor evaluation in patients with Parkinson’s Disease. They use quantitative measures calculated from 3D acceleration data and show a substantial concordance (>69.6%) of the accelerometric scores with the MDS-UPDRS ratings. However, the accelerometric kinetic tremor measures poorly associated with the given MDS-UPDRS scores. This is not surprising as it is not trivial to measure tremor from acceleration data in the presence of a prominent movement. The paper is well written and there potentially is a merit in establishing a quantitative way to eliminate human error. However, the methods are not very rigorous as the classification into different scoring ranges are done using a threshold determined from healthy subjects in a different environment. A discussion on why only surgical candidates were considered is also lacking. This would benefit more if performed in the testing performed during outpatient visits and would also provide patients with a wider range of tremor profile. Finally, use of or at least a discussion on more rigorous statistical classification methods for classifying the tremor into different score ranges would be useful fort he reader.

Author Response

In this paper, the authors propose a quantitative way of determining the MDS-UPDRS for tremor evaluation in patients with Parkinson’s Disease. They use quantitative measures calculated from 3D acceleration data and show a substantial concordance (>69.6%) of the accelerometric scores with the MDS-UPDRS ratings. However, the accelerometric kinetic tremor measures poorly associated with the given MDS-UPDRS scores. This is not surprising as it is not trivial to measure tremor from acceleration data in the presence of a prominent movement. The paper is well written and there potentially is a merit in establishing a quantitative way to eliminate human error. However, the methods are not very rigorous as the classification into different scoring ranges are done using a threshold determined from healthy subjects in a different environment. A discussion on why only surgical candidates were considered is also lacking. This would benefit more if performed in the testing performed during outpatient visits and would also provide patients with a wider range of tremor profile. Finally, use of or at least a discussion on more rigorous statistical classification methods for classifying the tremor into different score ranges would be useful for the reader.

The authors thank the reviewer for thoroughly assessing our revised manuscript and for providing critical comments. We are aware of the differences in measurement environment between the patient group and the healthy group. As there were no options for this research to be performed in healthy persons undergoing surgical procedures similar to the patient group, we had to accept the differences in measurement environment. We hope the reviewer agrees with us that it is not ethical to let the healthy participants undergo the same rigorous surgical steps (e.g. mounting a frame to the person’s head, drilling a hole in the skull) for the sake of performing noninvasive tremor measurements using accelerometry under the same circumstances of the participating patients. As described in the second paragraph Methods section 2.3, both the healthy and patient participants were measured in supine position. The healthy measurements used in our pilot study should suffice to demonstrate our method on translating the MDS-UPDRS tremor tests to a method based on accelerometry measurements.

We agree with the reviewer that our reason for focusing on neurosurgical PD patients should be mentioned in the discussion. As PD treatment decisions during awake thalamotomy and STN-DBS surgery, e.g. permanent lead position, currently depend in part on subjective MDS-UPDRS tremor assessments, this study focused on neurosurgical PD patients. We have added this clarification in section 4.1 of the Discussion. In our ongoing research, we also perform our accelerometry-based method at the out-patient clinic, which has indeed resulted in an even wider range of tremor scores than was presented in the current pilot study. These out-patient clinic results will be used in a future paper.

We also agree with the reviewer on the need for a description of other possible classification methods in tremor research in our discussion section. We have therefore added an explanation on this in the second paragraph in section 4.5 of the discussion.

Reviewer 2 Report

On page 2, the authors state that the Unified Parkinson Disease Rating Scale (UPDRS) is the best scale for assessing tremor in Parkinson disease. However, this has never been demonstrated. In fact, Pinter and colleagues found that the Fahn-Tolosa-Marin scale was actually better.1 Therefore, it is incorrect to think that the UPDRS is the gold standard.

The main problem in the operating room is the natural variability (within subject variability) of Parkinson tremor, particularly the rest tremor. This problem is not overcome by transducers. This has been discussed extensively in a recent review by Elble and Ondo.2

Also on page 2, the authors state that accelerometers are particularly well suited to measure tremor in the hand. This is actually untrue for reasons discussed by Elble and Ondo.

It is often stated that accelerometry is more objective than rating scales. However, there is plenty of subjectivity and user variability when using transducers. There is user variability in how the transducers are attached to the limb, how long tremor is recorded, how many times tremor is recorded, how to decide when recording a satisfactory or unsatisfactory, variability in task performance and instructions, etc.

The biggest problem with accelerometry is that the signal is contaminated by gravitational artifact and that the acceleration recorded by the accelerometer is a function of the distance of the accelerometer from the axis of rotation.2 The axis of rotation can vary greatly from moment to moment. For example, tremor may begin in the fingers and then spread to the wrist, elbow and shoulder. This is particularly true of severe tremors in patients undergoing surgery for tremor.

I assume that the authors use the autoregressive method of spectral analysis because their recordings were short (10 seconds). This needs to be clarified.

A big problem with this study is that the authors simply assume that spectral power in the 4-6 Hz band is tremor. This is a convenient assumption that is often false. Tremor cannot be assumed to be any power within the 4 to 6 Hz band. One must have objective criteria for assessing whether a spectral peak is present. If there is no spectral peak, there is no tremor.

Similarly, using an amplitude threshold above control levels is a poor way of detecting tremor because one cannot assume that all spectral power above control levels is tremor. This is particularly true of kinetic tremor. Previous authors have used some test of rhythmicity to establish the presence of or absence of tremor. This is analogous to demonstrating a statistically significant spectral peak. In my opinion, the authors’ approach to determining when tremor is present is a major weakness of this paper.

The authors’ approach to tremor quantification is not new. In fact, it is unclear what if anything is new in this paper.

Like most studies of this type, the authors show that an accelerometer can be used to quantify tremor and produce ratings that closely match clinician ratings. However, the authors don’t address the important question of whether it is worth the effort to use a transducer rather than straightforward clinical ratings. Previous studies have shown no advantage of one over the other from the standpoint of detecting clinical change that exceeds random variability. The precision of transducers is mitigated by the within subject variability of tremor. The authors believe that clinicians will eventually prefer transducers over rating scales. There is currently no reason to believe that this will happen, particularly in the operating room.

References

  1. Pinter D, Forjaz MJ, Martinez-Martin P, et al. Which Scale Best Detects Treatment Response of Tremor in Parkinsonism? J Parkinsons Dis 2020;10(1):275-282.
  2. Elble RJ, Ondo W. Tremor rating scales and laboratory tools for assessing tremor. J Neurol Sci 2022;435:120202.

Author Response

The authors’ approach to tremor quantification is not new. In fact, it is unclear what if anything is new in this paper.

The authors thank the reviewer for reading our revised manuscript and for providing extensive feedback. First, we would like to underline that our approach of translating the standardized criteria of MDS-UPDRS tremor tests to a scoring algorithm based on accelerometric measurements has, to our knowledge, not been done before. Previous studies have mainly focused on tremor characteristics derived from sensor data that correlate well with clinical scales. However, our approach focusses on applying the same scoring criteria used for clinical assessment to accelerometric tremor data, and comparing the calculated scores to the clinical scores. Also, thresholds calculated based on healthy control data were implemented in the scoring algorithm, which has not been done before. Moreover, our main outcome measure, which was the area under the curve of the power in the 4-6Hz frequency range, is another novelty of our manuscript.

On page 2, the authors state that the Unified Parkinson Disease Rating Scale (UPDRS) is the best scale for assessing tremor in Parkinson disease. However, this has never been demonstrated. In fact, Pinter and colleagues found that the Fahn-Tolosa-Marin scale was actually better.1 Therefore, it is incorrect to think that the UPDRS is the gold standard.

The authors agree with the reviewer that it is best to reformulate the term “gold standard”. We realize that the MDS-UPDRS may not be the gold standard, but more like the most frequently used and best validated clinical scale in PD. We have made this adjustment in the first sentence of the abstract and in the second paragraph of the Introduction.

The main problem in the operating room is the natural variability (within subject variability) of Parkinson tremor, particularly the rest tremor. This problem is not overcome by transducers. This has been discussed extensively in a recent review by Elble and Ondo.2

Also on page 2, the authors state that accelerometers are particularly well suited to measure tremor in the hand. This is actually untrue for reasons discussed by Elble and Ondo.

We agree with the reviewer that transducer-based measurements will and can currently not replace clinical assessments. As stated by Elble and Ondo, transducers like accelerometers can complement clinical assessment to quantify tremor severity. We have added this to section 4.5 of the Discussion. Transducers can hopefully form a meaningful addition by improving the inter- and intra-rater variability of clinical assessments. In our ongoing research, we are studying the inter-rater variability of clinical ratings and test-retest reliability of our accelerometry-based method, as stated in section 4.5 of the Discussion.

It is often stated that accelerometry is more objective than rating scales. However, there is plenty of subjectivity and user variability when using transducers. There is user variability in how the transducers are attached to the limb, how long tremor is recorded, how many times tremor is recorded, how to decide when recording a satisfactory or unsatisfactory, variability in task performance and instructions, etc.

We agree that the use of transducers falls prone to the possible subjectivity and user variability that any instrument operated by a human faces. This is why we have performed our measurements according to a strict protocol, ensuring placement of the sensors as described in our Method section and instructions according to the described MDS-UPDRS tremor tests. The recording time of the measurements was pre-set in our measurement program and the protocol was followed strictly when performing the measurements, preventing variability in how long and how many times tremor was recorded. Decisions on when a tremor measurement is performed satisfactorily will face the subjectivity of the rater whether the assessment is clinical or transducer-based. In our pilot study, the rater was a PD expert (T.v.L.).

The biggest problem with accelerometry is that the signal is contaminated by gravitational artifact and that the acceleration recorded by the accelerometer is a function of the distance of the accelerometer from the axis of rotation.2 The axis of rotation can vary greatly from moment to moment. For example, tremor may begin in the fingers and then spread to the wrist, elbow and shoulder. This is particularly true of severe tremors in patients undergoing surgery for tremor.

The authors realize that raw accelerometric data is contaminated by the earth’s gravitation. By applying a non-causal second-order high pass Butterworth filter with a cutoff frequency of 0.5Hz, we have minimized these artifacts. We have added to the first paragraph of Methods section 2.4 that this filter setting suppresses gravitational acceleration effects.

As the placement of the sensor and the axis of rotation are indeed critical for correctly performing accelerometric measurements, we have followed our exact protocol for accurate placement of the sensors and consistent performance of the tremor tests. As stated by Elble and Ondo, triaxial accelerometers will provide valid measures of hand tremor, if the axis of rotation is relatively constant. In their example, the sensor was located on the dorsum of the hand, which is not exactly the same, but close to our sensor placement.

I assume that the authors use the autoregressive method of spectral analysis because their recordings were short (10 seconds). This needs to be clarified.

As described in section 2.5 of the Methods, the periodogram PSD estimate was calculated to determine the power-based outcome measure PAUCtremor. The time length of the tremor recordings are not the main issue when calculating the periodogram, as it is more important that the sampling frequency is high enough (at least two times the Nyquist frequency). As the frequency range of our interest was below 10Hz and the sampling frequency of the accelerometer was 200Hz, we hope the reviewer agrees with us that this should suffice.

A big problem with this study is that the authors simply assume that spectral power in the 4-6 Hz band is tremor. This is a convenient assumption that is often false. Tremor cannot be assumed to be any power within the 4 to 6 Hz band. One must have objective criteria for assessing whether a spectral peak is present. If there is no spectral peak, there is no tremor.

We are aware that not all power in the 4-6Hz frequency range is caused by tremor. This is the main reason we included the data of healthy persons, to correct for activity in this frequency band that is present when a person does not suffer from any range of Parkinsonian tremor. Moreover, we introduced an outcome measure that took al power between 4Hz and 6Hz into account, by calculating the area under the curve of the periodogram for this frequency range (PAUCtremor). A dominant spectral peak that could be identified upon visual or automatic inspection of the periodogram, would also result in a substantially higher value of PAUCtremor. We have added this explanation in paragraph 5 of section 4.4 in the Discussion.

Similarly, using an amplitude threshold above control levels is a poor way of detecting tremor because one cannot assume that all spectral power above control levels is tremor. This is particularly true of kinetic tremor. Previous authors have used some test of rhythmicity to establish the presence of or absence of tremor. This is analogous to demonstrating a statistically significant spectral peak. In my opinion, the authors’ approach to determining when tremor is present is a major weakness of this paper.

We agree that kinetic tremor is especially hard to quantify. In our approach, we have attempted to correct for the acceleration caused by movement of the arm during kinetic tremor tests by applying a high-pass filter of 3Hz, as described in the second paragraph of Methods section 2.4.

Like most studies of this type, the authors show that an accelerometer can be used to quantify tremor and produce ratings that closely match clinician ratings. However, the authors don’t address the important question of whether it is worth the effort to use a transducer rather than straightforward clinical ratings. Previous studies have shown no advantage of one over the other from the standpoint of detecting clinical change that exceeds random variability. The precision of transducers is mitigated by the within subject variability of tremor. The authors believe that clinicians will eventually prefer transducers over rating scales. There is currently no reason to believe that this will happen, particularly in the operating room.

It is not our intent to show that clinical assessments can or should be replaced by transducer-based methods. As stated in the first paragraph of Discussion section 4.5, we see the use of objective measurement tools as an addition to the currently used clinical PD scales. Hopefully, objective methods will help to improve the inter- and intra-rater variability of clinical assessments. In order to investigate this, we are currently continuing our research on accelerometric measurements in movement disorder patients to, amongst others, determine reproducibility and test-retest reliability of our approach, as described in the first paragraph of Discussion section 4.5.

References

  1. Pinter D, Forjaz MJ, Martinez-Martin P, et al. Which Scale Best Detects Treatment Response of Tremor in Parkinsonism? J Parkinsons Dis 2020;10(1):275-282.
  2. Elble RJ, Ondo W. Tremor rating scales and laboratory tools for assessing tremor. J Neurol Sci 2022;435:120202.

This manuscript is a resubmission of an earlier submission. The following is a list of the peer review reports and author responses from that submission.

Round 1

Reviewer 1 Report

The submitted manuscript describes an interesting study that aims to use accelerometry to convert the MDS-UPDRS tremor tests into an objective method for assessing tremor severity. 
The MDS-UPDRS is the gold standard for tremor in Parkinson's disease, but it, like all other traditional clinical tests and rating scales, has some limitations. It is critical to continue developing technology-based objective measures to assess and monitor disease symptoms in order to improve patient care. 
Although the manuscript is interesting, there are some points that are unclear:

1) This study appears to be a case control study as presented. However, the author did not say whether the healthy controls and PD patients were matched.

2) What medication state were the patients in when the assessments were conducted?

3) It is not only necessary for the algorithm to correlate with clinical assessments. It is also necessary that the results they provide are simple to interpret and meaningful from a clinical standpoint. Please comment on the usefulness and clinical applicability of the developed objective outcome.

Author Response

The submitted manuscript describes an interesting study that aims to use accelerometry to convert the MDS-UPDRS tremor tests into an objective method for assessing tremor severity. The MDS-UPDRS is the gold standard for tremor in Parkinson's disease, but it, like all other traditional clinical tests and rating scales, has some limitations. It is critical to continue developing technology-based objective measures to assess and monitor disease symptoms in order to improve patient care. Although the manuscript is interesting, there are some points that are unclear:

1) This study appears to be a case control study as presented. However, the author did not say whether the healthy controls and PD patients were matched.

The authors thank the reviewer for reading our manuscript and for providing thoughtful comments. As described in the Results and Discussion (4.4 Inconsistencies) section of our paper, indeed, the participants were not matched. The main purpose of including healthy controls was to calculate thresholds for our algorithm. It was not our intention to match PD participants individually. Healthy participants’ data was used to benchmark our algorithm.

2) What medication state were the patients in when the assessments were conducted?

All measurements were performed in an off-medication state, after a washout period of at least twelve hours. We have added this information to the Abstract and to the Methods section (2.3 Measurements).

3) It is not only necessary for the algorithm to correlate with clinical assessments. It is also necessary that the results they provide are simple to interpret and meaningful from a clinical standpoint. Please comment on the usefulness and clinical applicability of the developed objective outcome.

We agree with the reviewer on the importance of usefulness and clinical applicability of our approach. We have added this aspect to the Discussion (4.1 Contributions).

Reviewer 2 Report

Dear Authors,

in this pilot study, you aimed to identify a reliable and rater-independent measurement method for monitoring tremor in Parkinson Disease, translating the MDS-UPDRS tremor tests to an objective method through accelerometry-based tremor amplitude scores. The topic could be interesting for the readership of this Journal, but I think a minor revision is necessary.

TITLE: In my opinion, for a better understanding of the paper's content, it would be appropriate to specify the acronym.

INTRODUCTION: In my opinion, in the Introduction authors should better investigate the limitations of current methods and clarify the study aim.

METHODS and RESULTS: I would suggest detailing the inclusion criteria, and better describing the study design.

DISCUSSION: This section is, in my opinion, well organized.

Author Response

Dear Authors,

in this pilot study, you aimed to identify a reliable and rater-independent measurement method for monitoring tremor in Parkinson Disease, translating the MDS-UPDRS tremor tests to an objective method through accelerometry-based tremor amplitude scores. The topic could be interesting for the readership of this Journal, but I think a minor revision is necessary.

TITLE: In my opinion, for a better understanding of the paper's content, it would be appropriate to specify the acronym.

The authors thank the reviewer for reading our manuscript and for providing feedback. We agree with the reviewer that it might be more clear to the reader to write the acronym “MDS-UPDRS’’ full out in the title. However, the title would be too lengthy if “Movement Disorder Society – Unified Parkinson’s Disease Rating Scale” was written in full. We have specified the acronym in the first sentence of the abstract. Therefore, we hope the reviewer will understand that we have chosen to retain the acronym “MDS-UPDRS” in the title.

INTRODUCTION: In my opinion, in the Introduction authors should better investigate the limitations of current methods and clarify the study aim.

We agree that these points should be clarified. As stated in paragraph 4 of the Introduction, the current gold standard (MDS-UPDRS) to assess parkinsonian tremor is associated with inherent subjectivity and reliance on experienced raters. So, PD treatment decisions, e.g. permanent lead position in neurosurgery, currently depend in part on subjective tremor assessments. Therefore, this study focuses on translating the MDS-UPDRS tremor tests to an objective method to rate tremor severity using accelerometry. This study also aims to show that these measurements can be performed in an intraoperative setting, allowing direct objective input for clinical decision-making. We have added this clarifying information to the Introduction, paragraph 4 and 8.

METHODS and RESULTS: I would suggest detailing the inclusion criteria, and better describing the study design.

As described in section 2.1 of the Materials and Methods, the inclusion criteria for all participants were: above the age of 18, and to be able and willing to adhere to the study. The inclusion criteria for patient participants was diagnosis of PD according to UK Brain Bank criteria, and undergoing unilateral thalamotomy or bilateral STN-DBS surgery. Inclusion criteria for the healthy group was absence of a current diagnosis of PD or any neurological disease. We have added that none of the participants were treated with tremor-inducing drugs during the study, in order to clarify that treatment with drugs that can induce tremor was not applicable for our study participants.

DISCUSSION: This section is, in my opinion, well organized.

The authors thank the reviewer for this positive feedback.

Reviewer 3 Report

I read the manuscript entitled: “Intraoperative Quantification of MDS-UPDRS Tremor 2 Measurements using 3D Accelerometry: A Pilot Study” I have the following comments and suggestions

Major points

Please explain in the introduction or methods section why it is important to have MDS-UPDRS-tremor scores and accelerometry intraoperatively. I feel that it would be much better to have those measurements in the laboratory under more controlled environment and using video-recordings. This is also important when comparing “healthy controls” and PD patients as they and the examiners were in different environments that can influence measurements. This should be acknowledge in the limitation section.

Video-recordings of tremor would simplify research as it can be used for intra and interrater comparisons once you have accelometry measurements. Do authors have these data?

It is unclear why this is a pilot study, are you conducting a larger study on this topic? Are these preliminary results? Otherwise the “pilot study” classification is not necessary and I would delete it.

I´m not aware that the MDS-UPDRS is a “gold standard” for parkinsonian tremor but rather the most commonly used tool, please check. Studies using accelerometry and interrater variability would be useful to better define the clinical tremor scoring and this should be a major aim of such studies, as tremor measurement by accelometry is rather the gold standard.

Minor points

In the introduction: What do you mean by heterogenic nature of PD? Do you mean heterogenic presentation? Or diverse pathogenesis? Please be more specific.

The use of the MDS-UPDRS requires written consent form from the MDS society, have the authors requested this one?

Please define Burg method.

Were “healthy controls” under treatment with drugs that can induce tremor (?) for example SSRIs, valproate, etc.

R2 I assume is coefficient of determination, please clarify in the “statistics” section

Author Response

I read the manuscript entitled: “Intraoperative Quantification of MDS-UPDRS Tremor Measurements using 3D Accelerometry: A Pilot Study” I have the following comments and suggestions.

Major points: 

Please explain in the introduction or methods section why it is important to have MDS-UPDRS-tremor scores and accelerometry intraoperatively. I feel that it would be much better to have those measurements in the laboratory under more controlled environment and using video-recordings. This is also important when comparing “healthy controls” and PD patients as they and the examiners were in different environments that can influence measurements. This should be acknowledged in the limitation section. 

We thank the reviewer for thoroughly assessing our manuscript and for providing critical comments and suggestions. As stated in the Introduction, the final position of the lesion or the DBS electrode during surgery is partly based on the clinical (MDS-UPDRS) findings. We agree that the measurements are easier and perhaps even better performed in a laboratory setting. However, we would like to stress that our objective measurements can be used in a real-life clinical setting, such as the operating theater, for the simple reason that treatment decisions on for example electrode position are made intraoperatively. With our study, we prove that a laboratory setting is not a pre-requisite for objectively measuring tremor. Since we understand the important point of the reviewer, we have added an explanation on the importance of performing these measurements intraoperatively in the Introduction, paragraph 4, 5 and 8. Considering the healthy controls being in a different setting compared to the patients, we fully agree that the differences in setting should be acknowledged in the limitation section. We have added this to the Discussion (4.4 Inconsistencies).

Video-recordings of tremor would simplify research as it can be used for intra and interrater comparisons once you have accelerometry measurements. Do authors have these data? 

Although we agree that video recordings are very useful for this type of research and for determining intra- and interrater variability, we do not have video recordings available. In our ongoing research, we have now added video recordings to the protocol, so for our upcoming study these video recordings will be available.

It is unclear why this is a pilot study, are you conducting a larger study on this topic? Are these preliminary results? Otherwise the “pilot study” classification is not necessary and I would delete it. 

Indeed, we are preparing a larger study, of which this manuscript is the preliminary chapter. As described in the Discussion (4.5 Future perspectives), intra- and inter-rater comparisons will be made in the future. We hope that the reviewer will understand that we therefore maintain the “pilot study” classification.

I´m not aware that the MDS-UPDRS is a “gold standard” for parkinsonian tremor but rather the most commonly used tool, please check. Studies using accelerometry and interrater variability would be useful to better define the clinical tremor scoring and this should be a major aim of such studies, as tremor measurement by accelerometry is rather the gold standard. 

We agree that the MDS-UPDRS is indeed the most commonly used tool for assessing parkinsonian tremor. There are several sources that claim that the MDS-UPDRS is the current gold standard for assessing disease state PD, as stated in the second paragraph of the Introduction. Therefore, we chose to retain the term “gold standard” in the manuscript.

Minor points: 

In the introduction: What do you mean by heterogenic nature of PD? Do you mean heterogenic presentation? Or diverse pathogenesis? Please be more specific. 

We have replaced the word “nature” by "presentation".

The use of the MDS-UPDRS requires written consent form from the MDS society, have the authors requested this one? 

The MDS-UPDRS scores were collected in a standard clinical setting. Based on the primary clinical use, permission from the MDS society was not obtained in advance. Nevertheless, we thank the reviewer for the suggestion and have issued a request for the use of the MDS-UPDRS for our future research. Permission for the use of the MDS-UPDRS was granted by the MDS society earlier this week.

Please define Burg method. 

We have added the definition of the Burg method in the Introduction, paragraph 7.

Were “healthy controls” under treatment with drugs that can induce tremor (?) for example SSRIs, valproate, etc. 

None of the participants were treated with tremor-inducing drugs during the study. We have added this information in the Methods section (2.1 Subjects).

R2 I assume is coefficient of determination, please clarify in the “statistics” section

We have added this clarification in Table 2.
